# Cardiovascular Consequences of Acute Kidney Injury: Treatment Options

**DOI:** 10.3390/biomedicines11092364

**Published:** 2023-08-24

**Authors:** Julija G. Voicehovska, Dace Trumpika, Vladimirs V. Voicehovskis, Eva Bormane, Inara Bušmane, Anda Grigane, Eva Moreino, Aivars Lejnieks

**Affiliations:** 1Department of Internal Diseases, Medical Faculty, Riga Stradins University, LV-1007 Riga, Latvia; dace.trumpika@rsu.lv (D.T.); vladimirs.voicehovskis@rsu.lv (V.V.V.); eva.moreino@rsu.lv (E.M.); aivars.lejnieks@rsu.lv (A.L.); 2Department of Kidney Diseases and Renal Replacement Therapy, Riga East Clinical University Hospital, LV-1038 Riga, Latvia; eva.bormane@aslimnica.lv (E.B.); inara.busmane@aslimnica.lv (I.B.); anda.grigane@aslimnica.lv (A.G.); 3Riga East Clinical University Hospital, LV-1038 Riga, Latvia

**Keywords:** acute kidney injury, cardiorenal syndrome, diuretic resistance, heart failure, hyperhydration, ultrafiltration

## Abstract

Soon after haemodialysis was introduced into clinical practice, a high risk of cardiac death was noted in end-stage renal disease. However, only in the last decade has it become clear that any renal injury, acute or chronic, is associated with high overall and cardiovascular lethality. The need for early recognition of kidney damage in cardiovascular pathology to assess risk and develop tactics for patient management contributed to the emergence of the concept of the “cardiorenal syndrome” (CRS). CRS is a pathophysiological disorder of the heart and kidneys in which acute or chronic dysfunction of one of these organs leads to acute or chronic dysfunction of the other. The beneficial effect of ultrafiltration as a component of renal replacement therapy (RRT) is due to the elimination of hyperhydration, which ultimately affects the improvement in cardiac contractile function. This review considers the theoretical background, current status of CRS, and future potential of RRT, focusing on the benefits of ultrafiltration as a therapeutic option.

## 1. Introduction

The relationship between heart and kidney disease has been studied by both cardiologists and nephrologists. Cardiorenal syndrome (CRS) appears to be an interference of heart and kidney diseases. Acute heart failure (AHF) is a syndrome defined as new onset (de novo) heart failure (HF) or worsening symptoms and signs from HF, primarily associated with systemic congestion [1]. In the presence of structural or functional cardiac dysfunction, one or more triggers may lead to AHF. In most patients with AHF, this occurs in the setting of pre-existing cardiac pathology, termed acute decompensated heart failure (ADHF). Unlike de novo AHF, patients with ADHF usually present with signs and symptoms of fluid congestion and retention (weight gain, exertional dyspnoea, orthopnoea, and resistant oedema) rather than the pulmonary oedema or cardiogenic shock characteristic of acute left ventricular (LV) systolic dysfunction. However, ADHF results from chronic, often unregulated, neurohumoral compensatory mechanisms that maintain the haemodynamic status quo despite deterioration of LV function. Decompensation develops when the balance swings toward fluid overload because the compensatory mechanisms are inadequate or do not function at all [2].

ADHF is one of the most common causes of hospitalization and represents a significant burden to the health care system. ADHF is associated with medication nonadherence, comorbidities, diet, modifiable risk factors, disease progression, and/or treatment failure, and all of these factors lead to high hospitalization rates. The data on annual hospitalizations due to HF exceed 1 million in the United States of America (USA) and Europe [3,4]. More than 90% of these hospitalizations were due to symptoms and signs of fluid accumulation (presumably ADHF). In addition, rehospitalization rates in the first three months after admission due to ADHF account for 30% in the United States and other countries, with up to one in four patients (24%) being re-hospitalized within 30 days and one in two patients (50%) within six months. Fluid accumulation in HF patients is not related to age and renal function, but is equally associated with worse outcomes [5]. Thus, data from the IMPACT-HF registry show that ADHF follows an alarming course and patients are hospitalized under more extreme conditions [2]. Despite significant advances in therapy and our understanding of the disease, treatment of ADHF is mainly symptomatic, based on decongestant agents and best adapted to the baseline haemodynamic state, without considering the underlying pathophysiological features. Greater individualization of treatment targeting aetiology and management after hospital discharge is urgently needed to improve long-term outcomes.

Because of multimorbidity, patients with ADHF invariably present with multiple comorbidities that often lead to urgent hospitalization. Demographic studies in ADHF patients have shown that cardiac comorbidities are common and can negatively impact treatment outcomes. Typical cardiac comorbidities include atrial fibrillation/flutter (30–46%), valvular heart disease (44%), and dilated cardiomyopathy (25%) [6]. Noncardiac comorbidities include renal dysfunction and diabetes mellitus. Observational studies in patients with ADHF show that 20–30% have renal dysfunction and 40% have diabetes mellitus [6,7].

Soon after the introduction of haemodialysis, a high risk of cardiac death was found in end-stage renal disease [8], but it was only in the last decade that it has become clear that any renal impairment, acute or chronic, is also associated with high overall and cardiovascular lethality. These data, obtained from large randomised trials [9,10], led to the concept of chronic kidney disease (CKD), developed under the auspices of the National Kidney Foundation, USA. The relationship between cardiovascular events and renal function persists after adjusting for the traditional cardiovascular risk factors, highlighting the independent defining role of renal function as such.

Standard treatment is usually conservative, mainly with intravenous loop diuretics. Resistance to diuretics often develops beyond heart failure (HF). One of the most effective methods of treating severe, drug-resistant CRS is the use of renal replacement therapy (RRT). The beneficial effect of ultrafiltration (UF) as a component of RRT is based on the elimination of hyperhydration, a reduction in the load on the heart due to the reduction in venous return, which ultimately has an effect on improving the contractile function of the heart. Here we aimed to present an analysis of the clinical data supporting the use of UF in CRS, with a particular focus on identifying clinical situations where such treatment options are beneficial.

## 2. Scope of the Problem

### 2.1. The Co-Dependency of Heart and Kidneys

Cardiorenal communication is important for controlling blood pressure, regulating the excretion of sodium and water by the kidneys, and ensuring adequate blood flow and oxygenation of tissues. Acute or chronic disease of one organ can lead to acute or chronic dysfunction of the other, and vice versa [11]. In HF patients, renal dysfunction often depends on many causes. In the case of salt and fluid retention and subsequent decompensation, it is crucial to find out the true cause [11,12]. Worsening of renal function (WRF) is defined as an absolute increase in serum creatinine of ≥0.3 mg/dL during hospitalisation and an increase of ≥25% from baseline [13]. WRF is the cause of complications in approximately 30% of HF hospitalisations and is associated with a longer length of stay, high readmission rates, and increased mortality, both short- and long-term. The physiological cardiorenal relationships have been presented as a haemodynamic model in which the kidneys control the volume of extracellular fluid by regulating the processes of sodium excretion and reabsorption, but the heart controls systemic haemodynamics. When any of the organs are damaged, the RAAS and sympathetic nervous system are activated, endothelial dysfunction and chronic systemic inflammation occur, and a vicious cycle is created in which the combination of cardiac and renal dysfunction leads to accelerated decline in the functioning of each of the organs, remodelling of the heart muscle, vascular wall, and renal tissue, and increased morbidity and mortality [14]. Thus, direct and indirect effects of each of the affected organs on each other can lead to the onset and persistence of combined cardiac and renal disease through complex neurohormonal feedback mechanisms [15]. This gridlock includes anaemia, which is present in many patients with CRS, whose incidence increases with the increase in NYHA functional class from HF and whose haemoglobin levels are inversely proportional to the size of the LV of the heart and the severity of LV hypertrophy [16,17].

An uncontrolled increase in urine output during diuretic therapy can lead to hypovolaemia and a decrease in preload, and the use of vasodilators can cause hypotension. In addition, non-steroidal anti-inflammatory drugs, cyclosporine, angiotensin converting enzyme inhibitors (ACE), and angiotensin II receptor blockers (ARA II) can also cause a decrease in renal blood flow.

### 2.2. Understanding Fluid Retention and Resistance to Diuretics

Diuretic resistance is the absence of effective decongestion (or, conversely, relief) despite adequate or increasing doses of diuretics. At the same time, a reduction in blood flow through the afferent artery and a reduction in sodium and chlorine concentrations in the distal nephron contribute to several pathophysiological effects that increase renin and angiotensin II production. Angiotensin II stimulates the adrenal cortex to produce aldosterone (ALDO). ALDO maintains fluid balance by supporting active sodium reabsorption in the kidney and colon, leading to fluid retention. On the contrary, an increase in sodium concentration in the distal nephron reduces the release of renin, angiotensin II, and ALDO. The increase in circulating blood volume (BCV) and LV pressure activate natriuretic peptide (NUP) synthesis. Normally, the main source of these peptides is the atrial tissue. When preload is increased, the synthesis of B-type atrial NUP (BNP) is initially accelerated; with sustained volume overload and remodelling of the ventricles, the synthesis of BNP increases markedly. NUPs have several effects on the nephron, including dilating afferent arterioles and constricting efferent arterioles, increasing intraglomerular pressure, decreasing mesangial tone, and improving effective renal filtration. In addition, ALDO production decreases due to inhibition of renin secretion secondary to a decrease in sodium reabsorption in the distal (convoluted) tubule of the kidney. Overall, the NUPs have an antagonistic effect on the RAAS. Volume overload or congestion in HF occurs due to increased filling pressures in the left and/or right ventricle. The role of the kidneys in the development of fluid overload and congestion is crucial even in the early stages of HF [18,19]. In HF patients, a decrease in effective BCV leads to sodium and water retention [20]. Initially, the compensatory effects of the neurohormonal control system led to adequate maintenance of perfusion pressure due to hyperhydration mechanisms and interstitial fluid accumulation, resulting in compensatory restoration of efficient BCV, improved venous blood return to the heart, and normalisation of filling pressure [21,22]. Increasing NUP activity in the early phase of HF helps to delay the development of cardiovascular changes and maladaptation. This could be achieved by increasing sodium excretion [23], preventing ALDO synthesis [24], increasing vasodilation [25], and preventing cell proliferation and inflammation [26].

Nevertheless, the effects of the RAAS remain without adequate countermeasures beyond sustained increased activity of vasopressor systems. As a result, excessive sodium and water retention, vasoconstriction, and volume overload occur [27].

Conventionally, two periods are distinguished in the development of stagnation: the haemodynamic period and the clinical manifestation period [28]. In the initial stage (up to several weeks), it is asymptomatic, and the pressure in the pulmonary artery and/or right heart is slightly elevated without progression [29]. A further increase in ventricular filling pressure may develop rapidly, with the transformation of haemodynamic stagnation into clinical stagnation and the appearance of symptoms and signs of ADHF. Such a clinical manifestation indicates that fluid retention is significant and should be considered accordingly. It is important to keep in mind that the increase in BCV in a patient with the development of ADHF is followed by a three-fold increase in interstitial volume, which is due to an uneven distribution of sodium [30]. A venous network is capable of absorbing up to 60–70% of BCV, mainly from the large visceral network [31]. In this context, if we consider isosmotic retention of sodium and water as the main cause of an increase in filling pressure of the heart, then weight gain of the patient could occur weeks before the manifestation of ADHF. The increase in BCV may be more than 100% in such patients (with an average increase of about 40%) [32]. However, persistent venous congestion several weeks before the deterioration of cardiac function suggests an independent and important association with the manifestation of ADHF [33,34]. In addition, several studies indicate that more than 50% of patients with ADHF did not have significant weight gain prior to worsening cardiac function, and weight gain in one month prior to hospital admission was less than 1 kg in these patients [35]. In this context, it is thought that redirection of circulating blood flow away from venous depots could lead to an increase in cardiac filling pressure, and these two mechanisms are often not mutually exclusive but complementary [36].

It is important to identify acute fluid diversion to detect general hyperhydration due to fluid retention in patients with the manifestation of ADHF, as this may influence the management of these patients.

Resistance to diuretics seems to indicate a worse prognosis in patients with HF. It is characterised by a decrease or absence of response to diuretics before improvement in symptoms associated with congestion [37]. It can be caused by a number of factors, including slowing of intestinal absorption of drugs associated with mucosal oedema, decreased renal blood flow, inadequate dosing of drugs, concomitant use of non-steroidal anti-inflammatory drugs that decrease the production of vasodilator and natriuretic prostaglandins, and increased salt intake. Diuretic resistance usually develops after the first doses of the drugs and is overcome more quickly by continuous infusion of furosemide than by its single administration followed by intravenous injection of thiazides. Combination therapy with diuretics requires careful monitoring as it can lead to excessive loss of sodium and potassium [38]. Long-term intravenous infusion of loop diuretics in resistant patients, as opposed to a single administration, may contribute to a more optimal and efficient delivery of the drug to the renal tubules, resulting in long-term inhibition of sodium reabsorption [39].

A Cochrane review compared a number of scientific papers on continuous and concurrent infusions of loop diuretics in patients with chronic heart failure (CHF): a better diuretic effect and shorter hospital stay were found in cases with prolonged administration [40]. In a summary of recommendations for the use of loop diuretics in patients with CHF depending on renal function, the author states that administration of the most effective doses of diuretics and salt restriction should be continued until an adequate response is achieved; in the case of a negative result, it is preferable to administer thiazide diuretics in combination with potassium-sparing drugs [41].

Aggressive diuretic therapy in this group of patients may lead to diuretic-induced hypovolaemia provoking renal failure, as has been reported in patients with acute CRS. Therefore, gradual achievement of a diuretic effect is preferable [42].

When congestion symptoms and the amount of circulating fluid increase, they are difficult to control with conventional therapeutic treatments. In patients resistant to diuretics, the UF method is used to reduce hypervolaemia with significant WRF and electrolyte disturbances [43,44].

### 2.3. Worsening of Renal Function and Cardiorenal Syndrome

Recently, due to the increase in the prevalence of cardiovascular pathologies, longer life expectancy of cardiac patients, and the use of interventional examination and treatment methods, the incidence of acute renal failure (ARF) has also increased [45]. The Acute Dialysis Quality Initiative Group (ADQI) has implemented the concept of acute kidney injury (AKI). A multilevel classification system, RIFLE (risk, injury, failure, loss of renal function, end-stage renal disease), was proposed for timely recognition, assessment of severity, and treatment of renal dysfunction, which was further modified by the Acute Kidney Injury Network expert group (AKIN) [46,47]. It is important to remember the possible iatrogenic causes for the development of CRS. An uncontrolled increase in urine output during diuretic therapy can lead to hypovolaemia and a decrease in preload, and the use of vasodilators can cause hypotension. In addition, non-steroidal anti-inflammatory drugs, cyclosporine, ACE inhibitors, and ARA II can also cause a decrease in renal blood flow. The need for early recognition of renal damage in cardiovascular pathology to assess risk and develop tactics for patient management contributed to the emergence of concepts such as “cardiorenal syndrome” [48], “cardiorenal anaemia syndrome” [49], and “cardiorenal continuum” [50]. Later, at the Acute Dialysis Quality Initiative (ADQI)-endorsed consensus meeting held in Venice in 2008, a classification was presented distinguishing five types of CRS [42]. A notable aspect was the recognition of the basic heterogeneity of CRS and the classification of its five main types based on the presence of acute/chronic HF and the primary/secondary occurrence of cardiac or renal damage in relation to each other and/or a significant increase in venous pressure.

CRS of the first type is a sudden, acute deterioration of the heart leading to acute kidney injury (AKI). It occurs in approximately 25% to 33% of patients admitted with acute decompensated HF [51]. Type 1 CRS occurs in 9–19% of cases in acute coronary syndrome [52]. ADHF is complicated by AKI in 24–45% of cases [53]. Clinical and laboratory manifestations of AKI usually develop within the first 4 days (50% of cases) or within 7 days (70–90% of cases) [54]. Because of the pathophysiological mechanisms involved in the development of CRS type 1, therapy for this condition aims to maintain and improve cardiac output and renal perfusion. High intra-abdominal pressure, and venous and renal congestion, in turn, require the use of diuretics and vasodilators in the early stages of treatment [54]. The purpose of diuretics is to reduce the volume of extracellular fluid at a rate that provides sufficient time for its passage from the interstitium to the vascular bed. However, the use of high doses of loop diuretics can be complicated by electrolyte imbalance, hypovolaemia, and hyperactivation of neurohormonal systems, which exacerbates AKI, and the phenomenon of diuretic inhibition and post diuretic sodium retention reduces susceptibility to diuretics. UF is a good alternative to loop diuretics for the correction of hypervolaemia in acute HF and WRF.

CRS of the second type is characterised by the presence of CHF in a patient leading to the development and progression of CKD over time. The incidence of CKD in patients with CHF can be 45–63.6% and is a poor predictor of cardiovascular death [55]. Therapeutic approaches should aim to eliminate and treat causes and/or diseases that lead to cardiovascular damage and WRF. Tight control of extracellular fluid and sodium balance is absolutely essential to prevent CRS type 2. At the same time, combinations of moderate doses of loop diuretics with other diuretics are preferred, as increasing the doses of loop diuretics is associated with adverse consequences due to the additional activation of neurohumoral mechanisms.

CRS of the third type or acute kidney syndrome includes primary and acute renal dysfunction due to acute glomerulonephritis, pyelonephritis, or acute tubular necrosis. AKI, in turn, causes HF, cardiac arrhythmias, and myocardial ischaemia. AKI occurs most frequently in ICU patients (35%) [56]. Timely elimination of electrolyte disturbances prevents arrhythmias and the associated haemodynamic changes.

CRS of the fourth type is characterised by CKD leading to CHF. The incidence of CKD has recently increased worldwide, reaching 10–15% [57]. One of the most recent systematic review and meta-analysis of observational studies estimating CKD prevalence in general populations reported a global CKD prevalence of between 11 and 13% with the majority being stage 3 [58].

Currently, the main causes of CKD are diabetes mellitus, arterial hypertension, atherosclerosis, and obesity, diseases that are most prevalent in developed countries. The risk of dying from cardiovascular disease is increased 10–20-fold in patients with CKD compared to patients without CKD [59]. The extremely high risk of cardiovascular complications, especially in the end stage, may be related to the combined effect of traditional and renal risk factors. To prevent volume overload and the development of HF, interdialytic weight gain should be minimised. Adequate sodium control (restricted diet and low dialysate) reduces the need for UF, the development of intradialytic hypotension, and episodes of repeated ischaemic “stunning” of the heart and brain.

CRS of the fifth type is a combination of cardiac and renal pathology as a result of acute injury, in which the dysfunction of one organ affects the function of another. Type 5 CRS occurs in septic patients or other critical conditions [60]; CKD and cardiac abnormalities in diabetes mellitus, cirrhosis, amyloidosis, and vasculitis might result in CRS of the type 5 as well. The mechanisms of development of this type of CRS are complex; the treatment today is to treat the underlying cause of the disease, generally using the same principles applied in CRS types 1 and 3. The use of intensive RRT in patients with sepsis has shown that blood purification improves the functional state of the myocardium and prevents cardiovascular complications.

In addition, a new definition was announced in 2022, renal tamponade or renal compression, to explain congestive nephropathy caused by limited space for renal expansion [61]. This could be another approach with possible further treatment options for HF by reducing intrarenal congestion [62].

### 2.4. Ultrafiltration

RRT methods, especially UF, haemodialysis (HD), haemofiltration (HF), and haemodiafiltration (HDF), are widely used in nephrology practice and in the management of critical conditions. One of the components of HD, HF, and HDF is UF, i.e., mechanical fluid removal, which ultimately leads to elimination of overhydration and more effective recovery of sensitivity to drug therapies. A number of studies have established the efficacy of UF as the method of choice in the treatment of severe CRS. Recently, a “hybrid” RRT technique has emerged in clinical practice, called sustained low-efficiency dialysis (SLED). SLED differs from intermittent HD in that it uses a lower blood flow rate, dialysate, and a longer duration of sessions. Sparse modes of this technique allow the maintenance of haemodynamic stability while performing clearance of low-molecular-weight, water-soluble substances. Because numerous studies have shown an association between fluid overload and morbidity and mortality, prevention of iatrogenic fluid overload through fluid restriction, avoidance of unnecessary fluid intake, and prudent use of diuretics is critical. However, many patients with oliguria and fluid overload respond only moderately to diuretics, especially if the renal impairment is severe or rapidly progressive. For patients who are not helped by conservative fluid management strategies, strategies that prevent fluid overload and shorten the duration of fluid overload are needed.

However, current guidelines mainly recommend UF in refractory HF when either high-dose intravenous loop diuretics or a combination of different diuretics are used.

The recommendations of the European Society of Cardiology Working Group (ESC), and the Heart Failure Association (HFA) on the diagnosis and management of acute and chronic HF, state that “there is no evidence that UF is preferred over loop diuretics as first-line therapy in patients with acute HF. At this time, routine use of UF is not recommended and should be limited to patients who do not respond to diuretic strategies” [63].

According to the National Institute for Health and Care Excellence clinical guideline on “Acute heart failure: diagnosis and management of acute heart failure in adults” [64], UF should not be routinely offered in patients with AHF, but it may be considered in patients with demonstrated diuretic resistance. Diuretic resistance is defined as the need to increase the dose beyond the previously recognized maximum dose or a dose approaching the maximum recommended daily dose without further improvement in diuresis.

The 2022 American Heart Association/American College of Cardiology/Heart Failure Society of America (AHA/ACC/HFSA) guideline on the management of HF is intended to provide patient-centred recommendations for clinicians and demonstrates that early initiation of UF after admission leads to a reduction in rehospitalizations. However, many aspects of UF, such as patient selection, fluid deprivation rate, venous access, avoidance of treatment-related complications, and cost, require further investigation [65].

The heterogeneity in the selection of the study population and UF indications and protocols, as well as the large differences in the pharmacological therapy used in the control group, may contribute to the interpretation of some conflicting results. Indeed, it must be emphasized that a more selective approach is needed when deciding on the most effective treatment modality for a given patient [65].

The co-dependency of heart and kidneys displayed in Figure 1.

## 3. Clinical Studies

Several clinical trials have investigated the role of UF in the treatment of patients with ADHF. Fluid deprivation and weight change have generally been used to determine the efficacy of UF therapy, while effects on renal function have generally been considered a safety measure. A number of secondary end points, such as HF-related readmissions and length of hospital stay, were also assessed. The most relevant studies are summarized in Table 1.

A prospective randomized controlled trial (CARRESS-HF) enrolled patients with AHF and signs of CRS and persistent congestion. The study compared UF with diuretic therapy in patients with ADHF and WRF. It was found that the use of a graded pharmacological therapeutic approach was preferable to a UF strategy in terms of safety of renal function, with weight loss comparable with both approaches. Importantly, UF compared with pharmacologic therapy was associated with more effective dehydration (greater weight loss and fluid removal), but also with increases in serum creatinine and neurohormonal activation; UF also was associated with a higher rate of adverse events [66,77]. However, in patients with AHF, the higher initial decongestion with UF had no association with WRF. In patients with an ejection fraction (EF) > 40%, UF was independently associated with WRF, and higher initial dehydration was associated with a higher rate of adverse clinical outcomes characterized by aberrant responses to decongestive therapy [78].

A meta-analysis of eight randomized controlled trials with 801 participants showed that UF increased fluid deprivation and weight loss and decreased rehospitalization and the risk of deterioration HF in congestive patients, making UF a safe and effective treatment option for volume overloaded patients with HF. UF was able to remove a greater net volume of fluid without increasing the risk of complications [67]. The use of extracorporeal therapies remains to be investigated, as experience with such modalities is insufficient. The disadvantages include the need for veno-venous access, and the associated cost of the device and related disposables.

A prospective randomized controlled clinical trial has shown that early UF is superior to diuretics in initiating treatment of volume overload in ADHF patients [68].

In addition, UF has been reported to slightly reduce readmission rates up to 30 days after an acute episode of decompensated HF and to contribute to greater weight loss when an individualized rather than a fixed frequency of UF is applied. For example, in the CUORE trial [69], the UF target was tailored to the clinical needs of each participant, not to exceed 75% of the achieved weight. In this study, patients treated with UF were reported to have fewer hospitalizations up to 6 months after hospital discharge. The AVOID-HF [70] and UNLOAD [71] studies also showed a lower readmission rate for HF after the use of UF devices; additionally, AVOID-HF [70] reported improved quality of life to the same extent as diuretic treatment. To date, there is no evidence that UF has a mortality advantage over intravenous diuretic therapy; however, the ongoing study PURE-HF (NCT03161158) re-evaluates this dilemma by examining CV mortality and myocardial infarction events 90 days after discharge in patients treated with specialized UF in addition to low-dose diuretics compared with intravenous diuretics alone. Moderate confidence evidence suggests that UF is likely to reduce HF-related rehospitalization in the long term, with a NNTB (number needed to treat for an additional benefit) value of 10. UF may reduce overall rehospitalization within 30 days or at the longest available follow-up [72].

A 2019 meta-analysis compared the effects of UF and diuretics on key clinical outcomes and showed that UF was associated with a significant reduction in rehospitalization rates. An increase in serum creatinine was observed in patients on high-dose diuretics. It is important to emphasize that WRF has been associated with greatly increased mortality HF [73].

A recent evidence-based review of the management of type 1 CRS concluded that the management of CRS type 1 is often challenging due to a variety of mechanisms leading to WRF and the lack of new treatment approaches targeting renal dysfunction in HF patients. UF appears to be complex and associated with high costs, so it is still not justified for CRS patients [74].

A multidisciplinary cardio-nephrology team approach has been recommended to achieve more effective treatment of patients with CRS [79,80].

Although UF is more effective in fluid removal compared with diuretics and reduces readmissions due to HF, it is not possible to prove that UF is superior to diuretics because of adverse events and mortality in the UF group. Treatment outcomes seem to vary because of the multiple causes and different aetiologies of patients [75].

The Early Continuous Ultrafiltration in Chinese patients with Congestive Heart Failure (EUC-CHF) trial was designed to evaluate UF efficacy in clinical parameter improvement without adverse events. Also, the trial is expected to establish a scoring system based on a Chinese population to guide early UF treatment in appropriate patients. EUC-CHF is one of the first controlled trials tailored to determine the benefit of UF within 24 h of hospital admission. The results are pending [76].

As is known, both renal function and renal filtration efficiency depend on the number of functionally active nephrons, as well as the permeability and surface area of the basement membrane of the glomeruli and Starling forces. The kidneys also possess an autoregulation mechanism by modulating vascular resistance, which helps to adjust blood flow. The studies demonstrate that measurements associated with vascular ageing (such as pulse pressure, pulse wave velocity, and ankle–brachial index) are associated with the development of CKD. CKD is associated with increased arterial stiffness (AS) and LVhypertrophy, two important risk factors for cardiovascular events. Conversely, AS and LV mass are interconnected and are associated with early signs of renal damage such as microalbuminuria [81,82].

## 4. Conclusions

Treatment of CRS aims to eliminate venous congestion and fluid retention, which could improve cardiorenal status. Diuretic agents appear to be the cornerstone of decongestive therapy in ADHF. In ADHF patients, the development of CRS during conventional treatment with decongestants impairs prognosis. Most studies confirm the efficacy and safety of UF in any type of CRS. UF is able to induce decongestion and remove a considerable volume of fluid; at the same time, the risk of side effects does not increase. The efficacy and safety of conventional therapies remain inadequate. Because of new, advanced insights into the pathophysiological mechanisms underlying ADHF, UF is becoming an attractive option for the treatment of ADHF and CRS.

The decision about UF application should be made according to the following stepped approach [66]:
In the case of congestion with volume overload consider loop diuretics, i/v root is preferable (at a dose of twice daily as an oral dose).Assess efficacy of diuretics by monitoring urine output and/or weight control.If no decongestive effect within the next 6 h—consider doubling the IV diuretic dose.If no decongestive effect within the next 6 h—UF should be considered.


Use of UF results in isosmotic volume loss without inducing electrolyte abnormalities. The efficacy and safety of conventional therapies remain inadequate, but UF is recommended to reduce volume overload in the case of diuretic resistance.

## 5. Future Directions

Joint multidisciplinary meetings of the cardio-nephrology team would be constructive for CRS patients and public health systems. Although research in this context has generally shown positive results, there are still some unclear questions and concerns, especially regarding the effects of UF on renal function and rehospitalization rates. Further studies are needed to investigate the side effects associated with UF procedures, assess the potential benefits, and correctly identify the patient populations that might benefit most from early UF treatment.

## Figures and Tables

**Figure 1 biomedicines-11-02364-f001:**
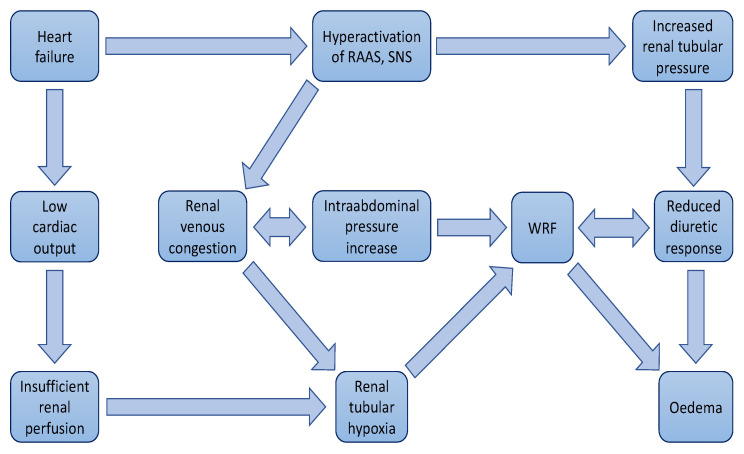
The co-dependency of heart and kidneys.

**Table 1 biomedicines-11-02364-t001:** Selected studies on the role of UF in the treatment of patients with ADHF when compared to medical therapy (clinician-based diuretic protocol).

First Author, Year	Study Type	Primary Outcomes	Results
Bart et al., 2012 [66]	Multicentred, prospective, RCT CARRESS-HF study	Change in serum creatinine level and change in weight.	UF is associated with more adverse events, WRF, and no change in weight.
Wobbe et al., 2021 [67]	Meta-analysis (PRISMA protocol)	Fluid removal, weight loss, all-cause mortality, heart failure-related rehospitalization, or adverse events.	UF increases fluid removal and weight loss and reduces rehospitalization.
Hu et al., 2020 [68]	RCT	Weight loss and an increase in urine output on days 4 and 8 of treatment.	Early UF is superior to diuretics for volume overload treatment initiation for ADHF patients.
Marenzi et al., 2014 [69]	RCT The CUORE trial	Rehospitalizations for congestive HF during a 1-year follow-up.	Clinical stabilization and less rehospitalization.
Costanzo et al., 2016 [70]	RCT The AVOID-HF	Time to first HF event within 90 days after discharge from index hospitalization.	UF group trended toward a longer time to first HF event within 90 days and had fewer HF and cardiovascular events.
Costanzo et al., 2007 [71]	RCT The UNLOAD trial	Weight loss and dyspnoea assessment at 48 h after randomization.	UF safely produces greater weight and fluid loss than intravenous diuretics.
Srivastava et al., 2022 [72]	Systematic search, RCT	Mortality and rehospitalisation rates.	UF probably reduces heart failure-related rehospitalisation in the long term.
Shi et al., 2019 [73]	RCT meta-analysis	Weight change, length of hospital stays, rehospitalization for HF, mortality, change in serum creatinine, dialysis dependence, and adverse outcomes.	UF was associated with significant reduction in the rate of rehospitalization.
Ong et al., 2021 [74]	Systematic search, RCT and observation studies	Changes in renal function tests.	Pharmacological therapy is recommended as the first-line therapy, and UF should only be reserved in cases of refractory congestion.
Wang et al., 2021 [75]	Systematic review and meta-analysis	Heart failure rehospitalization, all-cause rehospitalization, and mortality.	Although UF is more effective in removing fluids than diuretics and decrease rehospitalization due to HF and all causes, there is not enough evidence to prove that UF is superior because of adverse events and mortality in the UF group.
Yang et al., 2019 [76]	Open-label, registry-based, prospective study (EUC-CHF)	Changes of weight loss and dyspnoea severity score.	EUC-CHF is one of the first controlled trials tailored to determine the benefit of UF with 24 h from hospital admission. Results are pending.

## Data Availability

Not applicable.

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
