# Peer review of "Cardiovascular Consequences of Acute Kidney Injury: Treatment Options"

_biomedicines, 2023, doi:10.3390/biomedicines11092364_

Round 1

Reviewer 1 Report

In the complex world of heart failure, understanding the underlying concepts is paramount. Let's embark on a journey to explore these intricate details, beginning with the terminology that often confounds the uninitiated.

Understanding Volume Overload:

Imagine a system where fluids are not where they are supposed to be. This is what happens in "volume overload," a condition often linked to "decompensated heart failure." The term "heterogeneity" refers to the variability in this fluid distribution. But what does this all mean? Let's break it down.

Clarifying the Terminology: To grasp these concepts, one must first understand the terms. Heterogeneity is the diversity or differences within a system. Volume overload is an excess of fluid in the body, often leading to heart failure. Decompensated heart failure is a stage where the heart can't pump enough blood to meet the body's needs.

Visual Representation: Picture a heart struggling to function, with fluids misplaced. Diagrams or charts can help illustrate this fluid distribution, making the concept more tangible.

Practical Implications: Why is this important? Understanding volume overload is vital for patient care and management. It's not just about knowing the terms; it's about applying this knowledge to real-life situations.

Ultrafiltration in Heart Failure:

Next, we delve into the world of ultrafiltration, a treatment method for heart failure.

Summarizing Key Findings: Research has been conducted, studies have been analyzed, and conclusions have been drawn. Ultrafiltration has been found to be both effective and safe, but what does this mean for the patient?

Patient Perspective: Imagine being a patient undergoing this treatment. How would it affect your daily life or your overall well-being? These are questions that need to be addressed.

Cardiorenal Syndrome Management:

Managing cardiorenal syndrome is like balancing on a tightrope. Let's explore the methods.

Comparison of Methods: Continuous infusion or bolus injection? Each has its advantages and disadvantages. A side-by-side comparison can help medical professionals choose the right approach.

Clinical Guidelines: What do the experts say? Referencing current guidelines or opinions can guide the management strategies.

Evidence-Based Review:

Evidence is the cornerstone of medical practice.

Highlighting Key Evidence: Summarizing the evidence that supports the management of cardiorenal syndrome type 1 makes it accessible to a broader audience, from medical professionals to curious readers.

Link to Full Review: For those hungry for more, a link or reference to a full review can provide in-depth information.

Use of Diuretics:

Diuretics are like tools in a toolbox, each with a specific purpose.

Explaining Combination Therapy: Why use thiazide-type and loop diuretic agents together? In what scenarios? These questions need clear answers.

Patient Considerations: What about side effects or other considerations? These must be discussed to provide a complete picture.

Monitoring Techniques:

Monitoring is like having a watchful eye on the patient's condition.

Describing the Technology: Intrathoracic impedance monitoring is a complex term, but what does it mean? How does it work, and what are its clinical applications?

Real-World Application: Real-life examples or case studies can bring this technology to life, showing how it has been used in practice.

Intermittent Haemodiafiltration:

This is a complex process, but it doesn't have to be incomprehensible.

Simplifying Concepts: Breaking down terms like BNP and inflammatory cytokines and explaining their relevance to heart failure can make them more digestible.

Visual Aids: Visual aids can further simplify the process, making it easier to understand.

Challenges of Cardiorenal Syndrome:

Every treatment has its challenges.

Detailing the Challenges: What are the specific challenges of managing cardiorenal syndrome, and how do treatments like extracorporeal ultrafiltration address them?

Patient-Centered Approach: How do these challenges affect patient care and outcomes? This perspective must not be overlooked.

Role of Multimodal Strategies:

Multimodal strategies are like a symphony of approaches working in harmony.

Defining Multimodal Strategies: What are they, and how can they be implemented? Clear definitions and examples are needed.

Implementation Guidance: Best practices for integrating these strategies into clinical practice can guide medical professionals in their work.

Epidemiology of Acute Renal Failure:

Finally, we must explore the broader context of acute renal failure, especially in relation to heart failure.

Expanding on the Topic: More details and context are needed to fully understand this complex relationship.

Relevance to Heart Failure: The connection between acute renal failure and heart failure must be made clear to readers, illuminating the bigger picture.

By addressing these specific areas, the authors can craft an article that is not only clear and relevant but also accessible and valuable. Whether for medical professionals seeking guidance or general readers interested in heart failure and related topics, this comprehensive approach ensures that the information resonates with all.

The authors could revise the manuscript to include more detail on the methods used in the studies they reviewed.

They could also discuss the limitations of the existing research.

Minor editing of English language required

Author Response

xc

Dear Editor and Reviewers,

We would like to thank you for taking the time to review the manuscript and for providing helpful and constructive feedback. Alterations have been made in view of the comments and we think that the manuscript is stronger for it. Below are our replies to the comments you have made and the details of any subsequent amendments:

We would like to thank the reviewer for their thoughtful and insightful comments.

Understanding Volume Overload:

Imagine a system where fluids are not where they are supposed to be. This is what happens in "volume overload," a condition often linked to "decompensated heart failure." The term "heterogeneity" refers to the variability in this fluid distribution. But what does this all mean? Let's break it down. Clarifying the Terminology: To grasp these concepts, one must first understand the terms. Heterogeneity is the diversity or differences within a system. Volume overload is an excess of fluid in the body, often leading to heart failure. Decompensated heart failure is a stage where the heart can't pump enough blood to meet the body's needs.

Response: The authors agree that clarification of the terminology would be extremely helpful to the reader. The manuscript has been amended to reflect this (line 137)

 Visual Representation: Picture a heart struggling to function, with fluids misplaced. Diagrams or charts can help illustrate this fluid distribution, making the concept more tangible.

Response: We appreciate this important comment. While diagrams or charts may be informative, our fear is that such illustrations are beyond the scope of this review

Practical Implications: Why is this important? Understanding volume overload is vital for patient care and management. It's not just about knowing the terms; it's about applying this knowledge to real-life situations.

Response: The principal aim of this review is to harmonize and critique the literature supporting clinical management CRS. The word count limit also prevents adequate comparison and discussion of the nuance volume overload as a pathological process.

Summarizing Key Findings: Research has been conducted, studies have been analyzed, and conclusions have been drawn. Ultrafiltration has been found to be both effective and safe, but what does this mean for the patient? Patient Perspective: Imagine being a patient undergoing this treatment. How would it affect your daily life or your overall well-being? These are questions that need to be addressed.

Response: The authors recognize the importance of the point being made by the reviewer The authors have amended the manuscript to highlight this point, namely (line 352, line 377).

Comparison of Methods: Continuous infusion or bolus injection? Each has its advantages and disadvantages. A side-by-side comparison can help medical professionals choose the right approach.

Response: We appreciate this important comment. While further review of the various RRT scenarios may be informative, our fear is that such discussion is beyond the scope of this review. Especially considering the word count limit and given that our primary aim was to harmonize UF in CRS literature only.

Clinical Guidelines: What do the experts say? Referencing current guidelines or opinions can guide the management strategies.

Response: The authors recognize the importance of the point being made by the reviewer. The manuscript has been amended accordingly, the most current references that reflect the data on current guidelines or opinions are available (line 257); also references 59-60 reflect the most recent recommendations of the opinion leaders.

Highlighting Key Evidence: Summarizing the evidence that supports the management of cardiorenal syndrome type 1 makes it accessible to a broader audience, from medical professionals to curious readers.

Response: We appreciate this important comment. The word count limit also prevents adequate description of CRS Type 1 management. However, we provide some the most necessary information within section 2.3.

Link to Full Review: For those hungry for more, a link or reference to a full review can provide in-depth information. 

Response: We appreciate this valuable comment. We believe that Table 1 provides all the necessary information.

Explaining Combination Therapy: Why use thiazide-type and loop diuretic agents together? In what scenarios? These questions need clear answers.

 Response: We appreciate this important comment. While further review of the various diuretic agents may be informative, our fear is that such discussion is beyond the scope of this review. Especially considering the word count limit and given that our primary aim was to provide the theoretical background, current status of CRS, and future potential of RRT, focusing on the benefits of the ultrafiltration as a therapeutic option.

Patient Considerations: What about side effects or other considerations? These must be discussed to provide a complete picture.

Response: The authors recognize the importance of the point being made by the reviewer The authors have amended the manuscript to highlight this point (line 352, line 377).. 

Monitoring is like having a watchful eye on the patient's condition. Describing the Technology: Intrathoracic impedance monitoring is a complex term, but what does it mean? How does it work, and what are its clinical applications?

Response: The reviewer makes an important point. However, the word count limit also prevents adequate comparison and discussion of the Intrathoracic impedance monitoring

Real-World Application: Real-life examples or case studies can bring this technology to life, showing how it has been used in practice.

Response: We appreciate this important comment. While further review of real-life examples may be informative, our fear is that such discussion is beyond the scope of this review. Especially considering the word count limit.

Simplifying Concepts: Breaking down terms like BNP and inflammatory cytokines and explaining their relevance to heart failure can make them more digestible. However, it is the authors opinion that information being provided is sufficient for a readers overall understanding.

Response: The authors agree that explaining some terms like BNP can make them more digestible. However, it is the authors opinion that information being provided is sufficient for a readers overall understanding.

Visual Aids: Visual aids can further simplify the process, making it easier to understand.

Response: We agree totally, that visual aids can be helpful, this is why Figure 1 was developed. We are to consider space limit, so there are no other figures and/or pictures. We could try to develop supplementary materials, however, we believe that such pictures will be out of the scope of this narrative review.

 Challenges of Cardiorenal Syndrome: Every treatment has its challenges. Detailing the Challenges: What are the specific challenges of managing cardiorenal syndrome, and how do treatments like extracorporeal ultrafiltration address them?

Response: We totally agree with the comment. However, the word count limit also prevents adequate comparison and discussion of the nuance of any difference.

Patient-Centered Approach: How do these challenges affect patient care and outcomes? This perspective must not be overlooked.

 Response: We appreciate this important comment. The word count limit also prevents adequate comparison and discussion of the nuance of any difference. However, this important topic has been covered by the manuscript (line 257, 377)

Role of Multimodal Strategies: Multimodal strategies are like a symphony of approaches working in harmony Defining Multimodal Strategies: What are they, and how can they be implemented? Clear definitions and examples are needed.

Response: We appreciate this important comment. While further review of the various multimodal strategies may be informative, our fear is that such discussion is beyond the scope of this review. Especially considering the word count limit and given that our primary aim was to provide the theoretical background, current status of CRS, and future potential of RRT, focusing on the benefits of the ultrafiltration as a therapeutic option.

Finally, we must explore the broader context of acute renal failure, especially in relation to heart failure. Expanding on the Topic: More details and context are needed to fully understand this complex relationship. Relevance to Heart Failure: The connection between acute renal failure and heart failure must be made clear to readers, illuminating the bigger picture.

Response: The authors agree that clarification would be extremely helpful to the reader. Unfortunately word limit doesn’t allow detailed clarification, however we believe that Figure 1 is helpful and convenient enough.

By addressing these specific areas, the authors can craft an article that is not only clear and relevant but also accessible and valuable. Whether for medical professionals seeking guidance or general readers interested in heart failure and related topics, this comprehensive approach ensures that the information resonates with all. The authors could revise the manuscript to include more detail on the methods used in the studies they reviewed. They could also discuss the limitations of the existing research.

 Response: We would like to thank you for taking the time to review the manuscript and for providing helpful and constructive feedback. Alterations have been made in view of the comments and we think that the manuscript is stronger for it.

Reviewer 2 Report

The review “Heart Consequences of Acute Kidney Injury: Treatment Options” is well written and fulfills its stated objective of providing theoretical knowledge and the current state of the cardiorenal syndrome and the potential use of renal replacement therapy in this setting. In general, it is appreciated that the review is well-written and structured and that it meets the aim and scope of this journal. Some points to consider are as follows:

1.-In line 78, there is a mistake in the word “heart”.

2.-In line 64, there is an incomplete cite (6).

3.- Please write the meaning of NUP in section 2.2.

4.- In section 2.3, please mention the other causes of CRS type 5 besides sepsis.

5.-In line 257, please add the most current references that support the data on the prevalence of CKD.

Author Response

Dear Editor and Reviewers,

We would like to thank you for taking the time to review the manuscript and for providing helpful and constructive feedback. Alterations have been made in view of the comments and we think that the manuscript is stronger for it. Below are our replies to the comments you have made and the details of any subsequent amendments:

REVIEWER#2:
We would like to thank the reviewer for their supportive comments and for recognising the value in the work we are currently undertaking.

1.-In line 78, there is a mistake in the word “heart”.

Response: The manuscript has been amended accordingly.

2.-In line 64, there is an incomplete cite (6).

Response: The manuscript has been amended accordingly.

3.- Please write the meaning of NUP in section 2.2.

Response: The manuscript has been amended accordingly.

4.- In section 2.3, please mention the other causes of CRS type 5 besides sepsis.

Response: The reviewer makes an important point regarding different causes of CRS type 5. The manuscript has been amended to inform readers about other critical conditions resulting in CRS Type 5.

5.-In line 257, please add the most current references that support the data on the prevalence of CKD.

Response: The authors recognise the importance of the point being made by the reviewer. The manuscript has been amended accordingly, the most current references that support the data on the prevalence of CKD have been added.